# Evolutionary and Expression Analysis of *MOV10* and *MOV10L1* Reveals Their Origin, Duplication and Divergence

**DOI:** 10.3390/ijms23147523

**Published:** 2022-07-07

**Authors:** Shuaiqi Yang, Xiangmin Zhang, Xianpeng Li, Xiu Yin, Lei Teng, Guangdong Ji, Hongyan Li

**Affiliations:** 1College of Marine Life Sciences, and Institute of Evolution & Marine Biodiversity, Ocean University of China, Qingdao 266003, China; ysqsksg@163.com (S.Y.); 17662131436@163.com (X.Z.); lixianpeng0826@163.com (X.L.); 17806228937@163.com (X.Y.); 2School of Basic Medicine, Qingdao University, Qingdao 266071, China; qtlei@qdu.edu.cn; 3Laboratory for Marine Biology and Biotechnology, Pilot National Laboratory for Marine Science and Technology (Qingdao), Qingdao 266003, China

**Keywords:** RNA helicase, *MOV10*, *MOV10L1*, evolution, expression

## Abstract

*MOV10* and *MOV10L1* both encode ATP-dependent RNA helicases. In mammals, MOV10 and MOV10L1 participate in various kinds of biological contexts, such as defense of RNA virus invasion, neuron system, germ cell and early development. However, *mov10* and *mov10l1* in zebrafish are obscure and the evolutionary relationships of *mov10* among different species remain unclear. In this study, we found MOV10 and MOV10L1 had some variations despite they possessed the conserved feature of RNA helicase, however, they may originate from a single ancestor although they shared limited homology. A single *MOV10L1* gene existed among all species, while *MOV10* gene experienced lineage-specific intra-chromosomal gene duplication in several species. Interestingly, the *mov10* gene expanded to three in zebrafish, which originating from a duplication by whole genome specific duplication of teleost lineage followed by a specific intra-chromosome tandem duplication. The *mov10* and *mov10l1* showed distinct expression profiles in early stages, however, in adult zebrafish, three *mov10* genes exhibited similar diverse expression patterns in almost all tissues. We also demonstrated *mov10* genes were upregulated upon virus challenge, highlighting they had redundant conserved roles in virus infection. These results provide valuable data for the evolution of *MOV10* and *MOV10L1* and they are important to the further functional exploration.

## 1. Introduction

RNA helicases are implicated in all steps of RNA metabolism. MOV10 (Moloney leukemia virus 10) and MOV10L1 (Mov10 like 1) are RNA helicase homologs implicated in wide variety of biological processes such as mRNA translation, miRNA-mediated post-transcription, piRNA biogenesis, and so on [1]. MOV10, first identified in cell lines derived from mouse strains in which Moloney leukemia virus (M-MuLV) was inserted into the germ line, was known as a protein that prevents M-MuLV infection in mice [2]. MOV10L1, also known as CHAMP (cardiac helicase activated by MEF2 protein), with high homology to MOV10, was initially identified as a protein that expressed in testis specifically [3].

*MOV10* and *MOV10L1* both encode ATP-dependent helicases, which are enzymes that mediate ATP-dependent unwinding of RNA duplexes thus promoting structural rearrangements of RNP complexes [4]. There are six superfamily helicases classified by sequences, structures and mechanisms [5]. Eukaryotic helicases exclusively belong to the SF1 and SF2 superfamilies, which share a conserved core composed of two tandem domains with RNA binding ability and ATPase activity [6,7]. SF1 helicases have a (aspartic acid (D)-glutamic acid (E)-alanine (A)-glycine (G)) DEAG motif, rather than the more common (aspartic acid (D)-glutamic acid (E)-alanine (A)-aspartic acid (D)/Histidine (H)) DEAD/H box, which is the symbol of the SF2 helicases. To date, a lot of the identified RNA helicases belong to the SF2, only a dozen of SF1 RNA helicases have been identified. Among them, *MOV10* and *MOV10L1* both encode SF1 RNA helicases, which involved in various kinds of biological contexts, such as defense of RNA virus invasion, early development, neuron system, germ cell [8,9].

MOV10 is a versatile RNA helicase implicated in wide RNA regulation network. MOV10 is a nucleus/cytoplasmic protein and ubiquitously expressed in almost all tissues. MOV10 normally localizes to the RNA processing bodies (PB) in the cytoplasm. MOV10 has been reported in few organisms. In Drosophila, CG6967 or dMOV10 is induced specifically during the mid-gut regeneration, and inhibits intestinal stem cell proliferation, thus regulates the termination of mid-gut degeneration [10]. Knockdown of maternal *mov10* in xenopus embryos led to gastrulation defects, likely resulting from the large scales changes in mRNA expression due to Mov10’s role in miRNA mediated regulation of maternal-zygotic transition. In addition, zygotic Mov10 is necessary for development of the central nervous system [9]. In mouse, the *Mov10* knockout one led to embryonic lethality, suggesting MOV10 is critical for early development. The heterozygous mouse has less dendritic complexity in hippocampal neurons, implying a role of MOV10 in the development of brain [11]. MOV10 is also a nucleocytoplasmic protein expressed in spermatogonia and a determinant of spermiogenesis [1]. In Hela cell, MOV10 could interact with argonaute proteins and localizes to mRNA-degrading cytoplasmic P bodies, participating in the RNAi pathway and mediating miRNA-guided mRNA cleavage [12]. Later, it is found that MOV10 has a dual role in regulating translation, which facilitates the translation of some RNAs, but also increases the expression of other RNAs when bound to FMRP [13]. MOV10 interacts with UPF1, the crucial component of the nonsense-mediated mRNA decay pathway, promotes mRNA degradation [14]. Accumulating evidences suggest the key role of MOV10 is antivirus activity against HIV1, Influenza, HBV and HCV virus invasion [15,16,17,18,19]. MOV10 primarily executes the antiviral responses by two different ways. First, MOV10 regulates antiviral gene expressions including IFN signal and autophagy pathways. Second, MOV10 participates RNA degradation via miRNA pathways [8,20].

Contrary to the diverse expression patterns and versatile roles of MOV10, numerous studies demonstrate that MOV10L1 exhibits a restricted germ cell-specific expression patterns, emerges as a conserved key regulator of piRNA biogenesis [21,22,23,24,25]. While drosophila armitage (armi), supposed to be the common ancestor of vertebrate MOV10 and MOV10L1, is another MOV10/MOV10L1 homolog. Armi interacts with piRNA pathway factor Yb, tethered to piRNA precursors transcript, thus facilitating its processing into primary piRNAs in drosophila ovaries [22]. In mouse, the specific expression in germ cells of MOV10L1 increases during the development of sperms. MOV10L1 interacts with HSPA2 and Piwi proteins, participates in piRNA-directed retrotransposons silencing, thus implicates in male fertility [21]. Conditional knockout of *MOV10L1* in mouse disrupted the RNA helicase domain. The *Mov10l1* mutant male mice were sterile owing to meiotic arrest before the pachytene stage. In addition, MOV10L1 is associated with piRNA ribonucleoprotein particles (piRNPs), the *Mov10l1* mutant expresses Piwi proteins but devoid of piRNA, implying MOV10L1 is involved in piRNA biogenesis [25]. Disruption of MOV10L1 helicase activity leads to loss of pre-pachytene piRNAs, resulting in retrotransposons activation and early meiotic arrest [21,25,26]. MOV10L1 acts upstream of Piwi proteins and is also required for the biogenesis of pachytene piRNAs, a poorly characterized piRNA class expressed in the pachytene stage of meiosis, which has unique function in the maintenance of post-meiotic genomic integrity [24]. MOV10L1 possesses RNA helicase activity and exhibits 5′-3′ directional RNA unwinding activity. MOV10L1 binds piRNA precursors and is essential to activate piRNA processing [23]. Very recently, a yama mutant, which harbors V229E substitution in the N-terminal region instead of the helicase domain, leads to meiotic arrest, loss of pachytene piRNA and male infertility. The results reveal an important role of the MOV10L1 in piRNA biogenesis during male germ cell development. This accumulating evidence suggest that MOV10L1 act as a major regulator of piRNA biogenesis, and both the N-terminal and helicase domain are required for the function of MOV10L1.

So far, the MOV10 and MOV10L1 have been identified in several species, while MOV10 and MOV10L1 members in teleost are undetermined, and the evolutionary relationships of MOV10 among metazoan taxa remain unclear. In this study, we first analyzed the molecular evolution of MOV10 and MOV10L1 in representative species of metazoan, then examined the expression profiles of *mov10* and *mov10l1* in early development and adult tissues as well as in response to virus challenge in zebrafish.

## 2. Results

### 2.1. Three mov10 Genes and mov10l1 in Zebrafish

The MOV10 and MOV10L1 of human were used to search the Mov10 and Mov10l1 in zebrafish genome by BLAST. As a result, a *mov10l1* gene and three *mov10* homologs, designated as *mov10a*, *mov10b.1* and *mov10b.2*, respectively, were identified. This is consistent with the nomenclature in NCBI database. Next, we obtained their cDNA by RT-PCR, suggesting the identified candidate genes are expressed in vivo. The ORF of *mov10a* coded for a protein of 1001 amino acids, with a molecular mass of ~113.92 kDa and a pI value of ~9.02. The ORF of *mov10b.1* encoded a 1013 aa protein, with a molecular mass of ~116.41 kDa and a pI of ~8.87. The predicted Mov10b.2 protein was quite similar to Mov10b.1, consisted of 1015 aa, with a molecular mass of ~116.51 kDa and a pI of ~8.53. The ORF of *mov10l1* coded for a protein of 1106 aa, with a molecular mass of ~122.61 kDa and a pI of ~5.89 (Table 1).

The sequence similarity and divergence between Mov10s (zebrafish Mov10a, Mov10b.1 and Mov10b.2) and Mov10l1 were shown in Figure 1. In specific, the Mov10s shared higher homology, ranging from 50.2% to 68.3%, while either of Mov10s and Mov10l1 shared lower homology but diverged greatly, ranging from 31.2% to 34.1% (Figure 1A,B). The sequence alignment revealed the higher homology among Mov10s (Figure 1C). The sequence alignment also showed that the c-terminal parts of the proteins are more conserved compared to the N-terminal part of the proteins (Figure 1C).

### 2.2. Lineage-Specific Gene Duplication of MOV10 in Multiple Species

Integrated MOV10 and MOV10L1 sequences in representative species from invertebrates to vertebrates, including fly, ascidian, amphioxus, lamprey, cartilaginous fishes, bony fishes, frog, chicken, mouse and human, were identified from NCBI and Ensembl databases by BLAST. Intriguingly, there were distinct MOV10 members in specific species, in contrast to only one MOV10L1 in all selected species. In specific, lamprey, cartilaginous fishes (Chondrichthyes), chicken, mouse and human possessed one MOV10, while as fly, frog and ascidian had two Mov10 members. Additionally, amphioxus contained three Mov10. In bony fishes (Osteichthyes), torafugu only had one single copy of Mov10, channel catfish and rainbow trout possessed two Mov10, the most peculiar one was zebrafish, which had three Mov10 (Table 2, Appendix A). More than one MOV10 members existed in several species, demonstrating possible gene duplication events during evolution.

To explore the evolutionary relationship of MOV10 and MOV10L1 in metazoan, the sequences obtained were first aligned by ClustalW algorithm for further phylogenetic analysis. We constructed phylogenetic trees of MOV10 and MOV10L1 of selected species by Maximum-Likelihood and Neighbor-Joining methods of MEGA 7 (Figure 2, Appendix A). The two constructed phylogenetic trees were consistent. In detail, All the MOV10L1 proteins clustered together, while MOV10 proteins formed another clade. In the MOV10 clade, the Mov10 proteins in the fly, ascidian, amphioxus, were clustered on the outermost side in turn, respectively, demonstrating that they were more primitive and conservative during the evolution process. It could be speculated the novel MOV10 members possibly derived from the species lineage-specific gene duplication. Four Mov10 of cartilaginous fish clubbed together indicating they had high homology compared to the counterpart of other species. All the bony fish Mov10 members were clustered into two branches: Mov10a and Mov10b. The torafugu Mov10 was clustered with the Mov10a of other bonny fishes. It could be inferred the late evolved Mov10b might be generated from teleost-specific genome duplication. Interestingly, zebrafish Mov10b.1 and Mov10b.2 formed a more closed sub-branch, indicating they shared high homology compared with Mov10b of the other teleost. We proposed that zebrafish Mov10b.1 and Mov10b.2 may be originated from zebrafish lineage-specific gene duplication. Overall, although the numbers of MOV10 diverged among species, MOV10L1 were only single in all the representative species.

### 2.3. Genomic Structure and Synteny of MOV10 Genes

We analyzed the gene structures of zebrafish mov10s and mov10l1 using TBtools. The exon-intron genomic structure of zebrafish mov10s and mov10l1 were shown (Figure 3). The three mov10s in zebrafish shared similar intron-exon structure consisting of 22 exons, while zebrafish mov10l1 possessed 21 exons. It also showed that zebrafish mov10l1 gene was more compact than the three mov10s of zebrafish, which have similar gene length.

To explore the synteny, we analyzed the genes surrounding *MOV10* and *MOV10L1* genes in human and zebrafish. While zebrafish *mov10a* located on chromosome 6, *mov10b.1*, *mov10b.2* and *mov10l1* were all mapped to chromosome 8, on which *mov10b.1* and *mov10b.2* were linked to each other. Further analysis of the surrounding genes revealed that the upstream and downstream genes of *mov10a* had their orthologs located near to the human *MOV10*. Similarly, the flanking genes of *mov10b.1* and *mov10b.2* could be found in the neighbor around human *MOV10*. The conserved synteny suggested that *mov10a*, *mov10b.1* and *mov10b.2* were co-orthologs of the human *MOV10*. However, zebrafish *mov10l1* had no synteny to the human *MOV10L1* gene (Figure 4) though the *MOV10L1* were well co-linearized among human, rat, mouse, chicken frog and some fishes (Appendix A). Thus, synteny analysis showed that *MOV10* was relatively conserved while *MOV10L1* was not.

### 2.4. Gene Duplication and Selective Pressure Analysis

The phylogenetic trees of MOV10 and MOV10L1 in different species and the synteny among the zebrafish *mov10s* and the human *MOV10* suggested that the intra-species gene duplication of MOV10 may occur in some species. To verify whether it’s happening in zebrafish, we mapped the Circos plot of the inter-chromosomal relationships by TBtools. In zebrafish, *mov10a* had the conserved synteny with *mov10b.1* as well as *mov10b.2* while *mov10l1* had no synteny with the three genes (Figure 5). This suggested that *mov10b* (either *mov10b.1* or *mov10b.2*) was originated from duplication of *mov10a* during evolution.

The value of the nonsynonymous substitutions per nonsynonymous site (Ka)/ the number of synonymous substitutions per synonymous site (Ks) ratios was used to explore the selection pressures influencing sequence divergence for the tandem duplication events of *MOV10* and *MOV10L1* (Table 3). The Ka/Ks ratios of zebrafish *mov10s* and *mov10l1* ranged from 0.18 to 0.35, indicating that the zebrafish *mov10s* and *mov10l1* may have experienced purifying selective pressure during evolution. The Ka/Ks ratio of *mov10b.2* and *mov10l1* was worked out but it of *mov10b.1* and *mov10l1* did not, reminded us that *mov10b.2* may be duplicated from *mov10a* first, then *mov10b.1* originated from a tandem duplication event of *mov10b.2* during evolution. In addition, the Ka/Ks ratios of *MOV10S* and *MOV10L1* between zebrafish and human or zebrafish and catfish were between 0.12 and 0.23, showed that the *MOV10* and *MOV10L1* genes may have undergone purifying selective pressure during the evolution of species. In conclusion, the above analyses showed that the *MOV10S* and *MOV10L1* were conservative during the evolutionary process, and its function may be redundant to some extent.

### 2.5. Domain and Motif Distribution Analysis

To elucidate the characteristic of Mov10s and Mov10l1, we used SMART and Pfam to predict the conserved domain Mov10s and Mov10l1 in zebrafish. All the four proteins shared two conserved RecA-like helicase core domains: AAA_11 (Domain I) and AAA_12 (Domain II), which located in the c-terminal region (Figure 6). AAA_11 is a ATPase domain associated with a varied cellular activities, AAA_12 is a characteristic domain of RNA helicases belonging to superfamily 1. Instead of the common DEAD/H box, SF1 helicases normally possessed a DEAG motif. Based on the characteristics of protein sequences, all Mov10s and Mov10l1 definitely belonged to SF1 helicases. Mov10l1 also possessed another S1-like RNA binding domain in the N-terminal, which were absent from all Mov10s (Figure 6).

We applied MEME online tools to analyze the motif structure of zebrafish Mov10s and Mov10l1. All Mov10 proteins shared similar motif structures, containing ten motifs from motif 1 to motif 10. While motif 2, 4, 5, 6, 7, 8, 9, 10 were presented in zebrafish Mov10l1, in which motif 1 and 3 were missing (Figure 7). Specifically, the motif 4 and 5 were in the AAA_11 domain, and the AAA_12 domain has the motif 7, 8, 9 and 10, while the motif 6 was shared by the two motifs. The DEAG motif, the signature feature sequence of SF1 helicases, was located in the motif 5. In addition, the other specific amino acid sequences that Mov10 had ubiquitously in other species were also present in these motifs (Figure 1C). Zebrafish Mov10s and Mov10l1 proteins had conserved functional domain and similar motifs as other species, suggesting their functions may be conserved.

### 2.6. Expression Pattern of mov10s and mov10l1 in Zebrafish

The gene expression is generally related to its function. Therefore, we detected the expression pattern of *mov10s* and *mov10l1* in early embryonic development and adult tissues of zebrafish. All the four genes had quite different expression patterns in early stages of zebrafish (Figure 8A). Under the same number of PCR cycles, *mov10a* had a relatively stable robust expression during all the embryonic development stages, both maternal and zygotic. The *mov10b.1* has no maternal expression, and the slightly significant expression begun to be observed from 24 hpf. However, *mov10b.2* had no transcript in the early embryos. *mov10l1* only had maternal but not zygotic expression. In conclusion, the two *mov10* genes and *mov10l1* gene were expressed in early stages, suggesting they were involved in early development of zebrafish.

We also measured the expression of *mov10s* and *mov10l1* in adult tissues of zebrafish. The *mov10a* had a relatively abundant expression in all the tissues with the highest expression in testis, followed by ovary. The expression level of *mov10b.2* was lower than that of *mov10b.1*, although they had similar expression patterns in tissues except ovary in which *mov10b.2* is undetectable. Compared to *mov10a*, they both had weak expression in eye and brain and the highest expression in intestine (Figure 8B). Different from the diverse expression patterns of *mov10s*, it was obvious that *mov10l1* was exclusively expressed in the testis and ovary, especially in the ovary. *mov10l1* was specifically expressed in germ cells resembling of the expression pattern of *mov10l1* in human and mouse, implying Mov10l1 in zebrafish had similar roles. The three zebrafish *mov10s* had similar but distinct expression patterns, indicating that the function of three Mov10s may be redundant but diverged to some extent.

### 2.7. Expression Profiles of mov10s and mov10l1 upon Virus Challenge

MOV10 has been proved to be antiviral player in human and mouse. To explore whether all the three Mov10 have antiviral properties in zebrafish, we conducted the following experiments. First, we tested and determined the lethal concentration of GCRV viruses. We injected the 9-month-old zebrafish with 30 microliters of the virus through the intraperitoneal injection, then the tissues were gutted, and total RNA was extracted at 24, 48, 96 h post injection.

Considering *mov10l1* specifically expressed in testis and ovary, so we only detected the expression of three *mov10* genes. The qRT-PCR results showed that the three *mov10* genes were upregulated upon GCRV virus challenge compared to the uninjected group though the level of upregulation were different at all the three time points upon virus infection (Figure 9A–I). When the virus invaded, the increase level of *mov10b.1* expression was the highest among the three *mov10*, especially in liver, a key immune organ. The results imply that all three *mov10* shared the redundant roles and were involved in virus defenses in zebrafish.

## 3. Discussion

MOV10 and MOV10L1, belonging to SF1 RNA helicases, are implicated in diverse biological processes [1,8,20]. MOV10 and MOV10L1 homologs have been identified in human, mouse, frog and fly. However, in the lower animals including teleost, cartilaginous fish, protochordates, agnatha and invertebrates, no definite orthologs determined.

Previously, MOV10 and MOV10L1 helicases supposed to be evolved from an common ancestor, drosophila Mov10/Mo10l1 homolog Armitage [2], while our phylogenetic tree proved that drosophila Armitage was the ortholog of Mov10l1. In addition, we did not obtain any homolog of MOV10 and MOV10L1 in nematode. ERI6/7, which share the same function as MOV10, has been proposed to be a MOV10 homolog [8], however, we found it was not an authentic homolog of MOV10 (data not shown). Interestingly, contrary to a single copy of *MOV10L1* in all the selected species, the numbers of *MOV10* differs among species, which implies that gene duplication events occurred for *MOV10*. All the tetrapod, except frog, possess only one *mov10* gene in each species, lamprey and cartilaginous fish also have a single *mov10* gene. In invertebrate lineage, such as fruit fly, ascidian, there are two *mov10* genes. While in teleost lineage, the number of *mov10* differs from one to three among different fishes. The phylogenetic analysis revealed that all the MOV10 family members were clubbed into one clade, it could be divided into eight subclades. Clearly, some of the Mov10 members clustered in a species-specific manner in several species, including fly, ascidian, amphioxus, frog. These results implies that the species lineage duplication occurred in these species. In addition, the *mov10* genes in each of these species are resided in the same chromosome, which suggesting that the duplication is intra-chromosomal duplication but not tandem duplication. In teleost species, we found Mov10a and Mov10b clustered into different subclades, suggesting that this duplication of *mov10* in teleost lineage due to the third rounds of WGD in teleost [27]. Zebrafish Mov10b.1 and Mov10b.2 formed a small clade, furthermore, these two genes are located to each other, which implies an extra tandem duplication occurred in the zebrafish lineage. These results are not consistent with the well-established two rounds (2R) of whole genome duplication (WGD) theory [28,29]. A lot of new genes are originated from gene duplication, which is important to the evolution of genome and genetic robustness [30].

Although *MOV10* and *MOV10L1* genes differ greatly in evolution among different species, Mov10 and Mov10l1 share many common features. They are similar in size and have the characteristic domain of SF1 helicases: two conserved RecA-like helicase core domains located in the c-terminal region, however, Mov10 and Mov10L1 also have variations to some extent. The zebrafish Mov10s and Mov10l1 share eight conserved motifs, in which the last seven corresponding to the conserved domain, while the 1st and 3nd motif are only existed in mov10 proteins. The 1st, 2nd and 3rd motif are not included in the characterized domain; therefore, it is difficult to predict the role of these motifs, which merits further exploration. In addition, the three Mov10 members in zebrafish are alkaline, while Mov10l1 in zebrafish is acidic with PI values less than 6. This implies that Mov10 and Mov10L1 have distinct roles in vivo although they share homology. As expected, the mov10l1 and mov10 homologs among species fall into two clades separately in the phylogenetic tree, though they may originate from the same ancestor. The divergence between mov10 and mov10l1 is consistent with that mov10 and mov10l1 implicated in distinct intracellular pathways. Mov10L1 displays a specific expression in germ cell and has a key regulatory role in piRNA biogenesis [21,23,24,31]. Contrarily, *MOV10* shows diverse expression pattern and versatile roles [1,8,9,10,20,32,33].

The prediction of the subcellular localization is important for the understanding of the mechanism of the proteins. The prediction of the putative localization of Mov10 and Mov10l1 in zebrafish showed that they may localize in nucleus, cytoplasm, mitochondrial and other organelles, suggesting Mov10 and Mov10l1 could display dynamic subcellular localization under different contexts. The cellular localizations of mouse and human MOV10 have previously been analyzed. Consistent with the prediction, MOV10 and MOV10L1 display versatile subcellular localization under different circumstances. Most research has shown that MOV10 accumulated in the cytoplasm, normally co-localized with the RNA processing body. MOV10 is restricted in the nuclei of several human cell strains and postnatal mouse brain [11,34,35]. While MOV10L1 is a nucleocytoplasmic protein in germ cells [1]. Drosophila Mov10L1 homolog, Armitage, known to shuttle between nuage (germline-specific membrane-less organelles) and mitochondria, facilitates stepwise RNA processing within these two compartments ovaries [36,37]. MOV10L1 could interact with PIWI, in which the piRNAs are processed and tethered, then the formed complexes are transported into the nucleus to exert TE silencing [38]. Therefore, we speculate that MOV10L1 also could shuttle between cytoplasm and nucleus. The MOV10 are predicted in the mitochondria with high possibilities especially for MOV10B.1 and MOV10B.2, however, both MOV10 hasn’t been found in any organelles before. This implies that the functions of MOV10 are far more complex. The localization of MOV10 and MOV10L1 in organelles and the roles depending on are worth of further exploration. The diverse cellular localization of MOV10 and MOV10L1 hint the possibility that they perform multiple and dynamic functions through different mechanisms in vivo. Despite of the conservation of zebrafish Mov10 proteins, the subcellular localizations of them show similar but different diversified patterns, implying they may have redundant but distinct functions. Detailed subcellular localization and the corresponding roles of MOV10 is intriguing and worth further investigation.

A striking different feature of *MOV10* and *MOV10L1* is the expression pattern. *MOV10S* exhibit a diverse expression pattern, while *MOV10L1* is specifically expressed in germ cells. In agreement, three *mov10* members share similar but slightly different expression patterns in adult zebrafish, they ubiquitously expressed in almost all the tissues except for in few ones. *mov10l1* is restricted on ovary/testis. The duplicated genes have three main fates, non-functionalization, sub-functionalization and neo-functionalization. Similar expression patterns of *mov10s* suggested that the duplicated *mov10s* might have redundant functions in adult zebrafish, whereas distinct expression of *mov10s* in certain tissues suggested their unique functions in these tissues. Surprisingly, *mov10* members show distinct expression profiles in early embryos: robust expression of *mov10a* in all tested stages and weak expression of *mov10b.1* from 24 hpf but no expression of *mov10b.2* in all the stages. As for *mov10l1*, the only maternal expression is observed in early stage. The diverged expression of *mov10s* and *mov10l1* in embryos implies they have different roles to be explored in future. One critical role of *MOV10* is antiviral activity in human and mouse, the infected cells upregulate the expression of MOV10 in response to virus and MOV10 could inhibit viral infection by several different strategies [17,18,39]. Similarly, we also demonstrated that the expression of three *mov10* genes are induced by GCRV in several immune organs, suggesting duplicated *mov10* may have redundant conserved role in virus defenses.

In summary, we performed the phylogenetic analysis of MOV10 and MOV10L1 in this study and found contrary to single *MOV10L1* genes among species, lineage-specific intra-chromosomally duplications and tandem duplication of *MOV10* occurred among species (Figure 10), suggesting the origin of independent and continuous duplication. Different from the specific expression of *MOV10L1*, *MOV10* genes exhibit similar but distinct expression patterns. We also demonstrated *MOV10* genes are upregulated by virus challenge, highlighting they have redundant conserved roles in virus infection.

## 4. Materials and Methods

### 4.1. Zebrafish Strain

Zebrafish AB strain that was purchased from China Zebrafish Resource Center (http://www.zfish.cn/, accessed on 12 September 2019), was cultured in the Haisheng Zebrafish Circulation Culture Tank at 28 °C following the standard protocol. Embryos were cultured in E3 medium.

### 4.2. Sequence Retrieval and Phylogenetics Analysis

To identify *mov10* and *mov10l1* genes in zebrafish, we searched GenBank (http://www.ncbi.nlm.nih.gov, accessed on 18 October 2021) and Ensembl (http://www.ensembl.org, accessed on 18 October 2021) using homo sapiens MOV10 and MOV10L1 as queries by local BLAST programs. To verify the accuracy of the retrieval candidate genes, we predicted the conserved domain of the predicted proteins with Smart program (http://smart.embl-heidelberg.de, accessed on 5 November 2021) and Pfam 35.0 (http://pfam.xfam.org/, accessed on 5 November 2021). Then, we predicted some other characteristics of these proteins. The predictive molecular weight and isoelectric point (PI) for the Mov10s and Mov10l1 proteins were calculated from Compute pI/Mw tool (https://web.expasy.org/compute_pi, accessed on 10 December 2021). The sequence similarity and divergence of them were align by MegAlign. In addition, the four proteins’ subcellular localization was studied using WoLF PSORT (https://www.genscript.com/wolf-psort.html, accessed on 10 December 2021).

To motif analysis, we submitted the four proteins to Multiple Em for Motif Elicitation (https://meme-suite.org/meme/tools/meme, accessed on 11 December 2021) in the neighbourhood of homology, in which the number of expected motifs was set to 10, and the rest parameters were all default values.

The intron-exon structures of the *mov10s* and *mov10l1* genes were generated by TBtools, Visualize Gene Structure (https://github.com/CJ-Chen/TBtools, accessed on 13 December 2021) [40]. In addition, the zebrafish genomic data was download from RefSeq: NCBI Reference Sequence Database (ftp://ftp.ncbi.nlm.nih.gov/genomes/refseq/, accessed on 18 December 2021).

The MOV10 and MOV10L1 protein sequences of human and mouse were used to BLAST the MOV10 and MOV10L1 protein homologs of other species with GenBank (http://www.ncbi.nlm.nih.gov, accessed on 18 October 2021) and Ensembl (http://www.ensembl.org, accessed on 18 October 2021). After that, we removed the redundant transcript sequence of the same gene, and then checked one by one to acquire all the MOV10 and MOV10L1 protein sequences. Multiple alignment of the MOV10 and MOV10L1 proteins of all species or the zebrafish were performed using ClustalW method in MegAlign V7.0.26. The Maximum-Likelihood and Neighbor-Joining method was used to construct the phylogenetic tree with a bootstrap of 1000 replicates and the evolutionary distances were computed using the JTT matrix-based method.

### 4.3. Synteny Analysis and Selective Pressure Analysis

The gene syntenic data of the genes of the zebrafish and human was collected from Genome Date Viewer (https://www.ncbi.nlm.nih.gov/genome/gdv, accessed on 27 December 2021) and Genomicus v100.01 (https://www.genomicus.bio.ens.psl.eu/genomicus-100.01/cgi-bin/search.pl, accessed on 27 December 2021). After that, we drawn the picture using the above information by the software IBS (http://ibs.biocuckoo.org/, accessed on 5 December 2021).

The zebrafish inter-chromosomal synteny analyses of the *mov10s* genes were generated by TBtools, Advanced Circos. In addition, the zebrafish genomic data was download from Ensembl Database (ftp://ftp.ensembl.org/pub/current_fasta/danio_rerio/, accessed on 5 December 2021). The transcript sequences of zebrafish, catfish and human, which were corresponding to the protein sequences in phylogenetic tree, were download from NCBI. In addition, the Ka/Ks ratios of them were generated by TBtools, Simple Ka/Ks Calculator (NG).

### 4.4. Virus Amplification, Adult Zebrafish Injection and Expression Analysis

The EPC cell strains were cultured with standard medium conditions at 28 °C, 5% CO_2_ following the standard protocol. First, the GCRV virus strain was inoculated into the EPC cells to expand the number of viruses. After that, the virus collected above was used to determine its lethal concentration. Finally, twelve 9-month-old adult zebrafish were inoculated with half the lethal concentration of virus, and the control group was injected with the standard medium. Twenty-four, forty-eight and ninety-six hours after injection, they were dissected to extract total RNA from specific tissues.

Twelve healthy zebrafish, which included six females and six males at the age of six months, were anaesthetized on ice. Then they were killed and twelve tissues in the order of eye, brain, gills, heart, liver, spleen, kidney, intestine, testis, and ovary, were sampled. Total RNA was extracted from the twelve tissues samples that were ground and preserved in RNAiso Plus (Takara, Shinjuku City, Tokyo) and purified using Total RNA Kit I (OMEGA Bio-Tek, Winooski, VT, USA). The cDNAs were reverse transcribed using HiScript III RT SuperMix for qPCR (+gDNA wiper) (Vazyme, Nanjing, China) as guided by the manufacturer’s instructions and the amount of RNA were 1 μg for each tissue. The specific primers (Appendix A) for the zebrafish *mov10s* and *mov10l1* genes were designed using Primer Premier 6.0 based on existing sequences from NCBI, and synthesized by the company (Sangon, Shanghai, China).

The RT-PCR analysis was conducted using the Taq Master Mix (Dye Plus) (Vazyme, Nanjing, China). Reaction conditions were: 94 °C for 30 s as stage 1, then 30 cycles of 94 °C for 30 s, 60 °C for 30 s, 72 °C for 30sas stage 2, and 72 °C for 7 min as stage 3. Nucleic acid electrophoresis was performed using 0.75% agarose gel. To ensure the accuracy and authenticity of experimental data, three times of the above experiments were repeated.

The qRT-PCR analysis was conducted using the ChamQ SYBR qPCR Master Mix (Low ROX Premixed) (Vazyme, Nanjing, China) on the ABI 7500 real-time PCR system (Applied Biosystem, Singapore). The reaction conditions was performed as described previously [41], and the figures of the relative expression levels of each gene were generated by the GraphPad Prism 7.

## Figures and Tables

**Figure 1 ijms-23-07523-f001:**
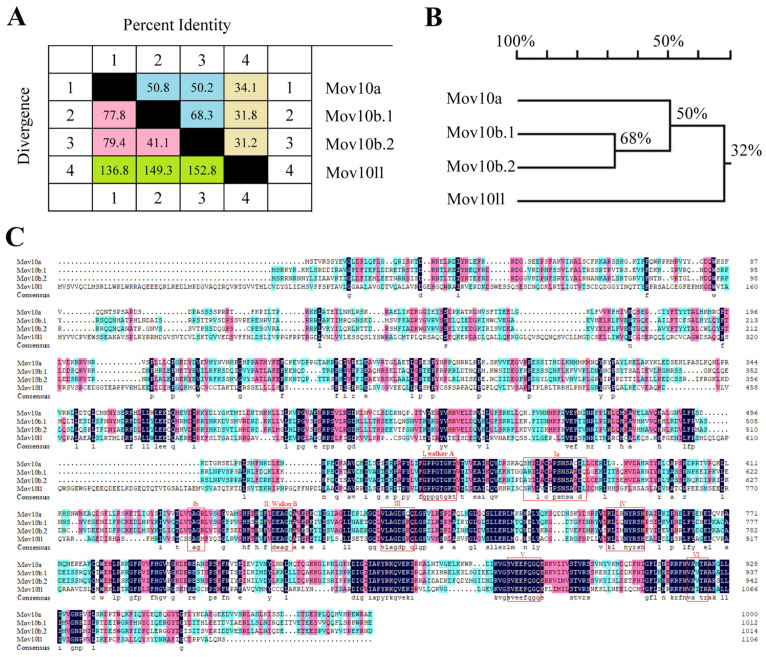
The protein sequences alignment of the Mov10s and Mov10l1 in zebrafish. (**A**) The percent identity and divergence of Mov10s and Mov10l1. The blue and pink areas, respectively, represent the percent identity and divergence between the three Mov10s, and the yellow and green areas show the percentage between either of Mov10s and Mov10l1. (**B**) The phylogenetic tree of Mov10s and Mov10l1. Numbers at each branches mean the percentage of their similarity. (**C**) The sequences alignment of Mov10s and Mov10l1. The dark blue area means 100% similarity; pale red, 75%; light blue, 50%; colorless, 0. The red boxes represents conserved sequences from previous studies.

**Figure 2 ijms-23-07523-f002:**
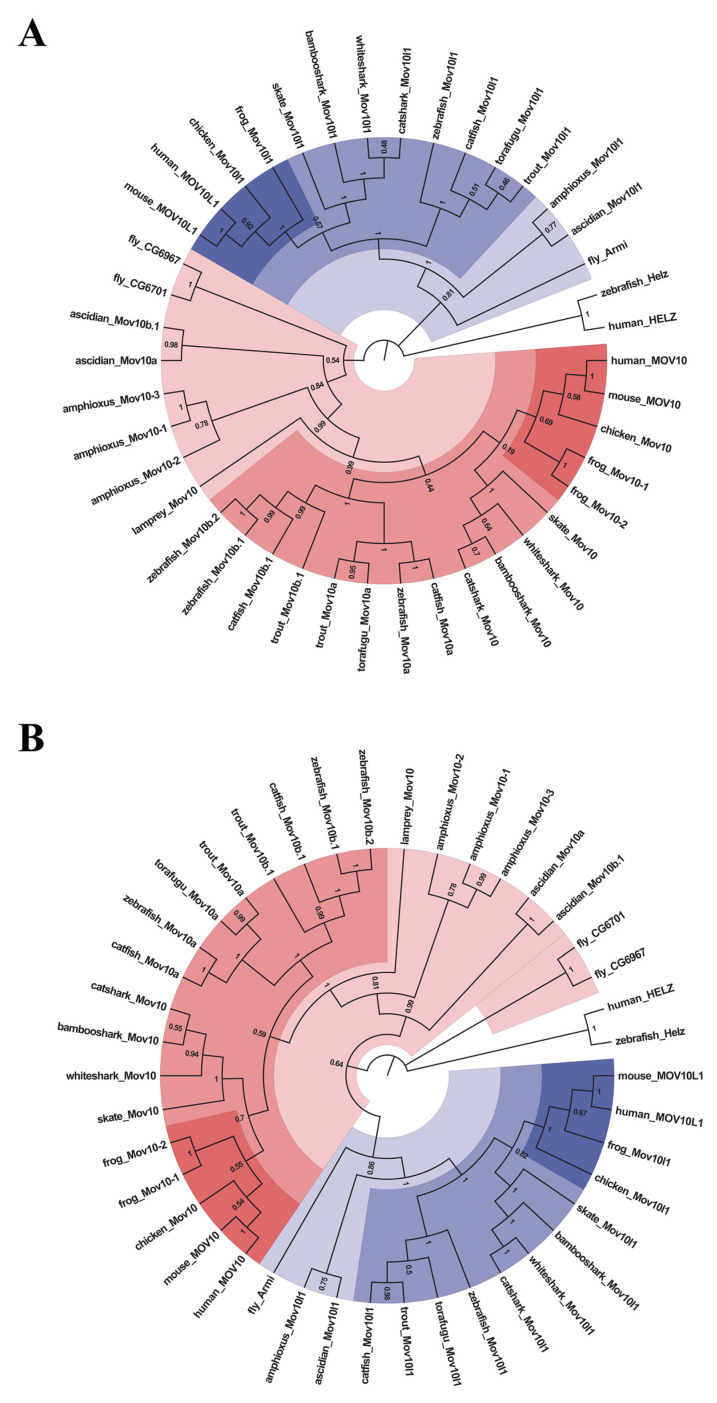
Phylogenetic tree analysis of the MOV10 and MOV10L1 in the selected species above. They are constructed by Maximum-Likelihood method (**A**) and Neighbor-Joining method (**B**). Different colored areas represent different evolutionary status, with darker colors indicating higher evolutionary status. Numbers at each branches mean the percentage of replicate trees in which the associated taxa clustered together in the bootstrap test (1000 replicates). Both trees are rooted with human and zebrafish helz.

**Figure 3 ijms-23-07523-f003:**
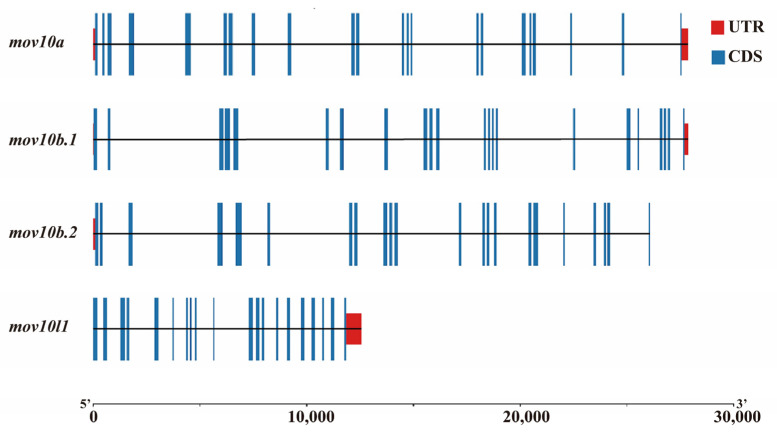
The intron-exon structures of the *mov10s* and *mov10l1* genes in zebrafish. The blue boxes denote the gene coding regions, the red boxes mean the untranslated region, and the lines represent the introns. The graduated line at the bottom shows the length of the genes.

**Figure 4 ijms-23-07523-f004:**
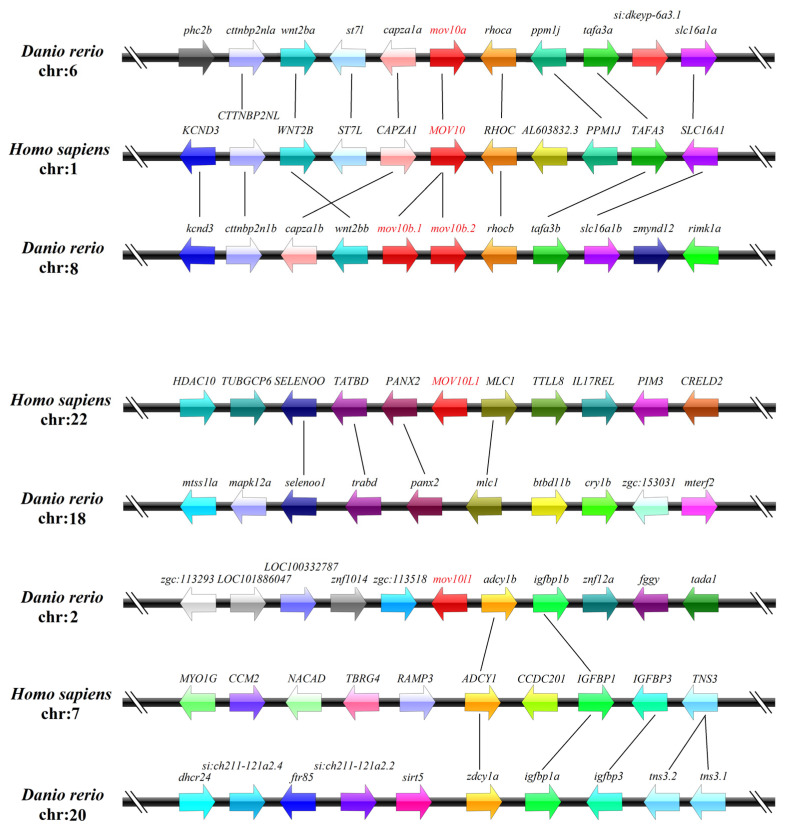
Synteny analyses of the *MOV10* and *MOV10L1* genes in human and zebrafish. The *MOV10* and *MOV10L1* genes are marked in red. The same color and the line between the genes of the two species mean they are homologous genes. The direction of the arrow represents the direction of the gene. The spatial distribution between different genes on the chromosome is indicated by bold black lines.

**Figure 5 ijms-23-07523-f005:**
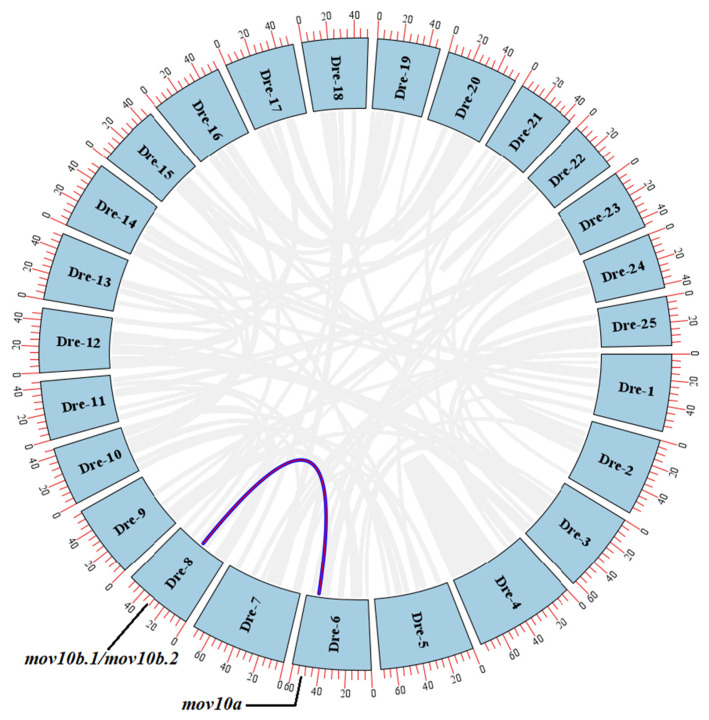
The circos plot of *mov10s* in zebrafish inter-chromosomal synteny analyses. The blue boxes represent the numbered zebrafish chromosomes. The outer red scales indicate the lengths of the chromosome sequences, with each scale representing 5 MB. The gray lines inside between the two chromosomes indicate the homologies between the genes, while the other colored lines indicated them between *mov10s*. Dre, *Danio rerio*.

**Figure 6 ijms-23-07523-f006:**
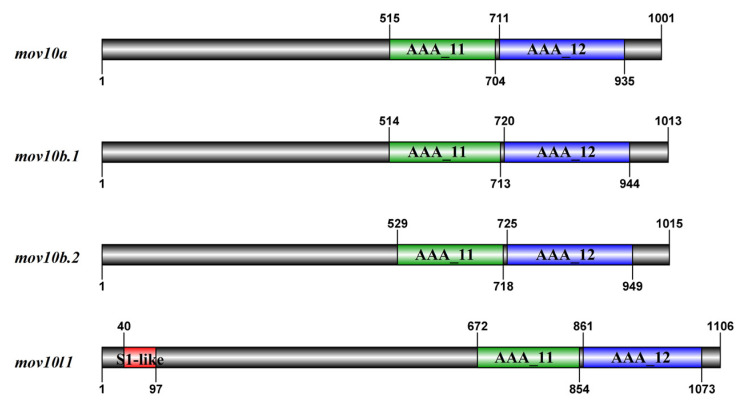
The schematic diagrams of domain organization in zebrafish Mov10s and Mov10l1. The bold black lines mean the amino acid sequence of the proteins. The higher squares represent the predicted domains of Mov10s and Mov10l1.

**Figure 7 ijms-23-07523-f007:**
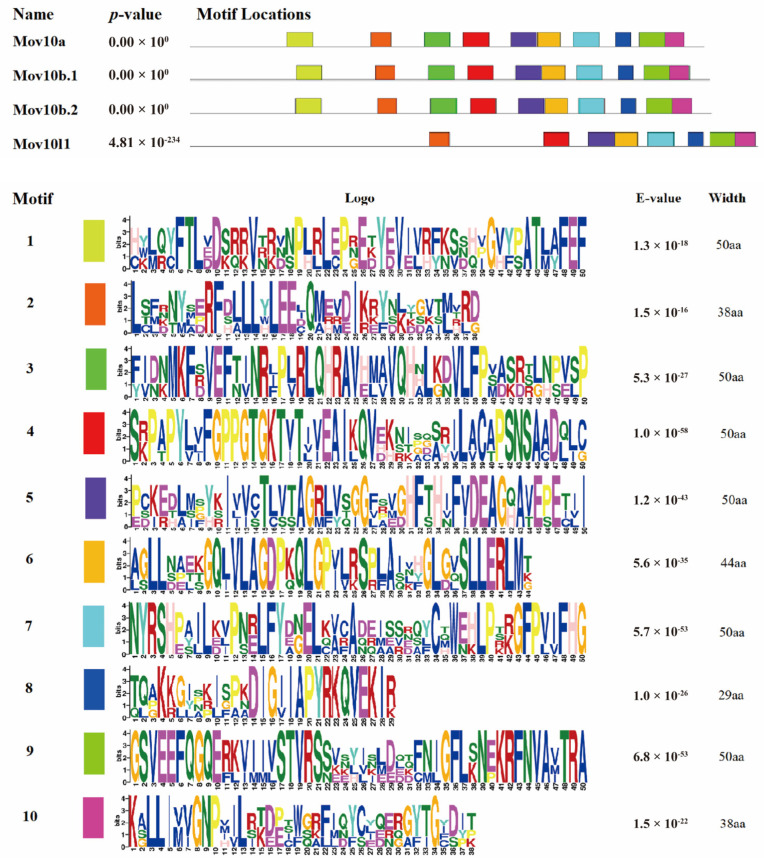
The schematic diagram of motifs of zebrafish Mov10s and Mov10l1 proteins. The motifs are arranged according to the position on the protein sequence, the letters in each motif are amino abbreviation. The size of the letter represents the saliency of the amino acid in the motif. The larger the letter, the higher the saliency, which is, the higher the frequency at which the amino acid appears in the same position in the same motif in different sequences.

**Figure 8 ijms-23-07523-f008:**
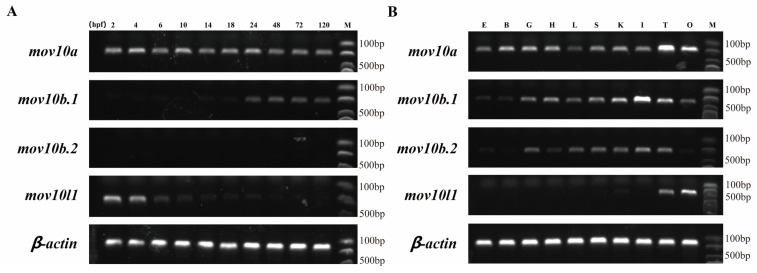
The Expression electrophoretogram of zebrafish *mov10s* and *mov10l1* genes. The transcript expression level of each *mov10s* and *mov10l1* gene was determined by RT-PCR in 10 embryonic development stages (**A**) and 10 tissues from six adult zebrafish (**B**). The number of PCR amplification cycles was 30. E, eye; B, brain; G, gills; H, heart; L, liver; S, spleen; K, kidney; I, intestine; T, testis; O, ovary; hpf: hour post fertilization; M, DNA Markers.

**Figure 9 ijms-23-07523-f009:**
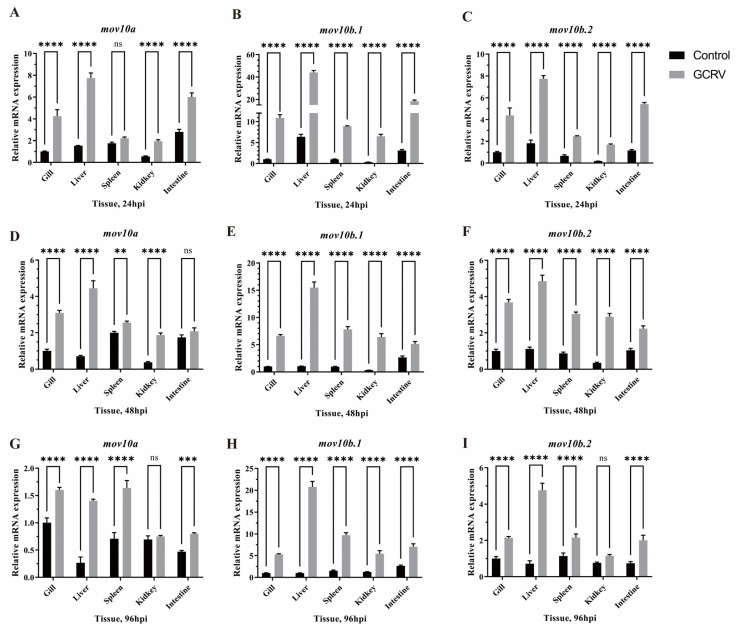
The antiviral properties of *mov10s* in various zebrafish tissues at different times upon virus infection. Compared to the control groups, the expression levels of *mov10a*, *mov10b.1* and *mov10b.2* at 24 hours (**A**–**C**), 48 hours (**D**–**F**), 96 hours (**G**–**I**) after GCRV virus injection. ** means *p* < 0.01; *** means *p* < 0.001; **** means *p* < 0.0001; ns, not significant. hpi, hours post injection.

**Figure 10 ijms-23-07523-f010:**
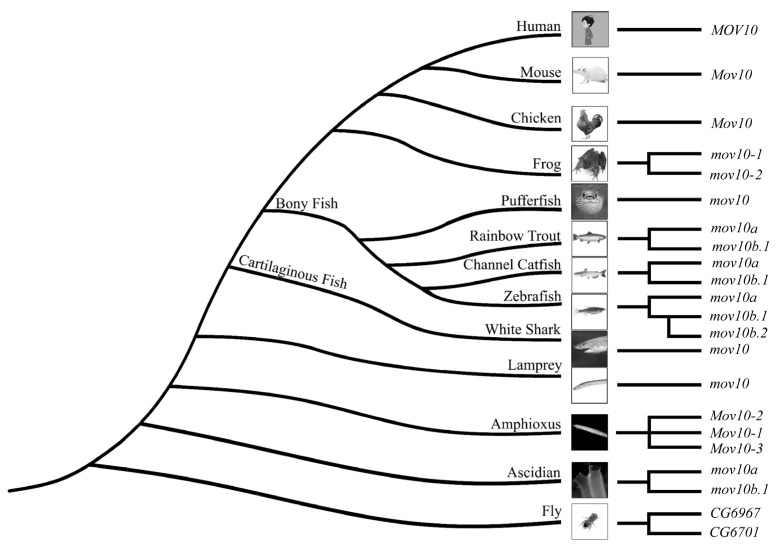
The lineage-specific intra-chromosomally duplications and tandem duplication of mov10 occurred among species.

**Table 1 ijms-23-07523-t001:** Summary of the features of the *mov10s* and *mov10l1* in zebrafish.

Gene	Chromosome	No. Exon	Coding Exon	CDS (bp)	Amino Acids (aa)	Molecular Weight (kDa)	pI	Localization
*mov10a*	6	22	22	3006	1001	113.92	9.02	C, N, P
*mov10b.1*	8	22	22	3042	1013	116.41	8.87	C, N, M
*mov10b.2*	8	22	22	3048	1015	116.51	8.53	C, N, M
*mov10l1*	8	21	20	3321	1106	122.61	5.89	C, N, M, E

Abbreviations for protein locations are: C, cytoplasm; N, nucleus; M, mitochondrion; E, extracellular; P, Integral membrane protein.

**Table 2 ijms-23-07523-t002:** Comparison of identified the *MOV10* and *MOV10L1* gene copy numbers across different representative species.

Class	Species	*MOV10*	*MOV10L1*
Mammalia	human	1	1
	mouse	1	1
Aves	chicken	1	1
Amphibia	frog	2 (1, 2)	1
Osteichthyes	torafugu	1	1
	catfish	2 (a, b.1)	1
	trout	2 (a, b.1)	1
	zebrafish	3 (a, b.1, b.2)	1
Chondrichthyes ^1^	shark	1	1
Cyclostomata	lamprey	1	1
Leptocardii	amphioxus	3 (1, 2, 3)	1
Ascidiacea	ascidian	2 (a, b.1)	1
Insecta	fly	2 *	1

^1^ The cartilaginous fishes (Chondrichthyes) include whiteshark (*Carcharodon carcharias*), bambooshark (*Chiloscyllium plagiosum*), skate (*Amblyraja radiata*), catshark (*Scyliorhinus canicula*). The * means that the two *mov10* genes of fly are CG6701 and CG6967.

**Table 3 ijms-23-07523-t003:** The Ka/Ks ratios of the *MOV10* and *MOV10L1* genes in zebrafish and other three representative species.

Seq_1	Seq_2	Ka	Ks	Ka/Ks
Dre-*mov10a*	Dre-*mov10b.1*	0.429010153	2.306444827	0.186004949
Dre-*mov10a*	Dre-*mov10b.2*	0.459220325	2.228487154	0.206068195
Dre-*mov10b.1*	Dre-*mov10b.2*	0.215764568	0.641967028	0.336099143
Dre-*mov10a*	Dre-*mov10l1*	0.73581865	2.163735682	0.34006864
Dre-*mov10b.1*	Dre-*mov10l1*	0.782334787	NaN	NaN
Dre-*mov10b.2*	Dre-*mov10l1*	0.779131356	3.864842108	0.201594615
Dre-*mov10a*	Hsa-*MOV10*	0.52853637	2.393945269	0.220780473
Dre-*mov10b.1*	Hsa-*MOV10*	0.558510073	2.654557187	0.2103967
Dre-*mov10b.2*	Hsa-*MOV10*	0.572195229	NaN	NaN
Dre-*mov10l1*	Hsa-*MOV10L1*	0.440656796	3.495808363	0.126052904
Dre-*mov10a*	Ipu-*mov10a*	0.20299231	1.431621356	0.141791898
Dre-*mov10b.1*	Ipu-*mov10b.1*	0.309622035	1.920636668	0.16120802
Dre-*mov10b.2*	Ipu-*mov10b.1*	0.300891219	2.060577589	0.146022756
Dre-*mov10l1*	Ipu-*mov10l1*	0.216627513	1.314532038	0.164794396

Abbreviations for the species names are: Dre, *Danio rerio*; Hsa, *Homo sapiens*; Ipu, *Ictalurus punctatus*. Ka: nonsynonymous substitution rate; Ks: synonymous substitution rate; Ka/Ks: the nonsynonymous substitutions per nonsynonymous site/the number of synonymous substitutions per synonymous site ratios; NaN: Not a Number.

## Data Availability

The transcript sequences and protein sequences of all species, which was needed for the research content in this paper, was searched and download from NCBI. The zebrafish genomic data was download from Ensembl Database (ftp://ftp.ensembl.org/pub/current_fasta/danio_rerio/, accessed on 5 December 2021; ftp://ftp.ensembl.org/pub/current_gff3/danio_rerio/, accessed on 5 December 2021) and RefSeq: NCBI Reference Sequence Database (ftp://ftp.ncbi.nlm.nih.gov/genomes/refseq/, accessed on 5 December 2021).

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
