# Peer review of "Evolutionary and Expression Analysis of MOV10 and MOV10L1 Reveals Their Origin, Duplication and Divergence"

_ijms, 2022, doi:10.3390/ijms23147523_

Round 1

Reviewer 1 Report

The manuscript ‘Evolutionary and Expression Analysis of MOV10 and MOV10L1 Reveals Their Origin, Duplication and Divergence’ by Yang et al. describes the evolutionary analysis of MOV10 & MOV10L1 proteins from selected metazoans. Experimental findings for the expression profiles of mov10 & mov10l1 genes in zebrafish were discussed in the early developmental stage, in adult tissues, and during the viral infection. Basically, the manuscript has extensively used bioinformatics tools to understand gene duplication, divergence, genomic integrity, and synteny of genes focusing MOV10 & MOV10L1 proteins in zebrafish. Though the manuscript is written well, however, it can be improved by adding more thoughts in the concluding remarks by projecting the future works to motivate young researchers.

The specific comments, which could help to improve the manuscript, are:

1. Introduction:

a.     Line 31: State some examples of biological processes.

b.     Line 105: Mov10 and Mov10l1 seems to be gene. If so, write in Italics. (also, in line 144,183, in fig5, 473). Throughout the manuscript check gene name & protein name and make them consistent.

2. Results:

a.     Line 120: Table 1 is showing gene information so their names should be written in italics

b.     Line 122: Figure 1 is showing sequence alignment. However, it is missing in the main text. Figure S2 should be merged with Figure 1. Pink is the correct representation of drawn color rather than Red. Figure 1, is shown with two segments, label them as A & B also edit the figure caption accordingly.

c.      Line 141&152: Which table or figure from supplementary information is being referred to?

d.     Line 171: Figure 2, is shown with two segments, label them as A & B also edit the figure caption accordingly as there is no Left & Right (only top & bottom). Present a high-resolution tree picture, as the text is not readable.

e.     Line 248: Figure 6, What do you mean by higher squares? How big are these AAA_11, AAA_12 domains? It will be better if sequence residues are labeled to show domain boundaries on amino acid sequence length (on the black line).

f.      Line 258: Conclusion word should be used in the right place. Better reframe it.

g.     Line 260: In Figure 7, protein names should be changed to Mov10a, Mov10b.1, and so on. Motif locations based on color are fine but starting sequence number should also be mentioned for each motif in each sequence.

h.     Line 278: Figure 8, In both A & B what is the last lane? Is it a marker? But, it doesn’t have any relevance in the explanation. 

3. Discussion:

a.     Line 338: “Genes are arisen” make it correct.

b.     Line 347: different should be replaced with difficult.

c.      Line 367: Drosophila-spelling check.

d.     Line 403: Lineage-spelling check.

4. Materials and Methods:

a.     Line 416: 28+1 C or 29 C.

b.     English can be improved by thorough reading. e.g. Line 447: Syntenic data……was collected. Wherever needed use “data were” rather ‘data was’(line 453,456 etc). even many places was/were not used appropriately.

c.      Many places words were not separated with space. Wherever needed add single space e.g. Veryrecently-line94, conditionsat-line 559, orderof 467, twelvetissue-line469, 30sas-478 etc,.

d.     Line 478: Than or Then?, Avoid starting a sentence with Then.

Author Response

Thanks.

Reviewer 2 Report

Yang et al. analyzed the evolutionary and expressional relationship of the MOV10 and MOV10L1. The manuscript is based on an in-silico study only. In my opinion, the manuscript lacks biological evidence.      

Major concerns

The expression pattern of mov10s (mov10a, mov10b.1, mov10b.2) upon viral infection is better analyzed in a time-dependent manner (for example day 1, day 3, and day 5 after viral injection).  

As mov10 prevents viral infection by multiple pathways including the activation of IFN signal, cell autophagy, and miRNA pathways. The authors should perform additional experiments to check the expression of potential markers of these pathways.

Minor concerns

There are several typo mistakes throughout the manuscript for example.

Line 467, anaesthetizedon

Line 469, twelvetissues

Line 309, Figure 9. The antiviral properties of mov10s and mov10l1 in various zebrafish tissues. Here, the mov10l1 expression is not examined, the authors should revise the figure title.

Author Response

Thanks.

Round 2

Reviewer 2 Report

In my opinion, the revised manuscript is suitable for publication in IJMS.